# Mechanical properties of lithic raw materials from Kazakhstan: Comparing chert, shale, and porphyry

Abay Namen[1,2☯¤]*, Radu Iovita[1,3], Klaus G. Nickel[4,5], Aristeidis Varis[1,6], Zhaken Taimagambetov[2,7], Patrick Schmidt[1,4☯]

1 Department of Early Prehistory and Quaternary Ecology, University of Tübingen, Tübingen, Germany,
2 Department of Archaeology, Ethnology and Museology, Faculty of History, Archaeology and Ethnology, Al-Farabi Kazakh National University, Almaty, Kazakhstan, 3 Center for the Study of Human Origins, Department of Anthropology, New York University, New York, New York, United States of America, 4 Applied Mineralogy, Department of Geosciences, Eberhard Karls University of Tübingen, Tübingen, Germany, 5 Competence Center Archaeometry – Baden-Württemberg, Eberhard Karls University of Tübingen, Tübingen, Germany, 6 Institute of Archaeological Sciences, Eberhard Karls University of Tübingen, Tübingen, Germany, 7 National Museum of the Republic of Kazakhstan, Nur-Sultan, Kazakhstan

☯ These authors contributed equally to this work.
¤ Current address: Department of Early Prehistory and Quaternary Ecology, University of Tübingen, Tübingen, Germany
* abay.namen@uni-tuebingen.de

**Data Availability Statement:** All relevant data are within the manuscript. Map data sources: Global Administrative areas (GADM), vector and raster

## Abstract

The study of lithic raw material quality has become one of the major interpretive tools to investigate the raw material selection behaviour and its influence to the knapping technology. In order to make objective assessments of raw material quality, we need to measure their mechanical properties (e.g., fracture resistance, hardness, modulus of elasticity). However, such comprehensive investigations are lacking for the Palaeolithic of Kazakhstan. In this work, we investigate geological and archaeological lithic raw material samples of chert, porphyry, and shale collected from the Inner Asian Mountain Corridor (henceforth IAMC). Selected samples of aforementioned rocks were tested by means of Vickers and Knoop indentation methods to determine the main aspect of their mechanical properties: their indentation fracture resistance (a value closely related to fracture toughness). These tests were complemented by traditional petrographic studies to characterise the mineralogical composition and evaluate the level of impurities that could have potentially affected the mechanical properties. The results show that materials, such as porphyry possess fracture toughness values that can be compared to those of chert. Previously, porphyry was thought to be of lower quality due to the anisotropic composition and coarse feldspar and quartz phenocrysts embedded in a silica rich matrix. However, our analysis suggests that different raw materials are not different in terms of indentation fracture resistance. This work also offers first insight into the quality of archaeological porphyry that was utilised as a primary raw material at various Upper Palaeolithic sites in the Inner Asian Mountain Corridor from 47–21 ka cal BP.

map data from Natural Earth (www.
naturalearthdata.com) and Shuttle Radar
Topography Mission (SRTM) Version 4 (https://
srtm.csi.cgiar.org/).

**Funding:** This project has received funding from
the European Research Council (ERC) under the
European Union's Horizon 2020 research and
innovation programme (grant agreement n˚
714842; PALAEOSILKROAD project). We
acknowledge support by the Open Access
Publishing Fund of the University of Tübingen. The
funders had no role in study design, data collection
and analysis, decision to publish, or preparation of
the manuscript.

**Competing interests:** The authors have declared
that no competing interests exist.

## 1 Introduction

In recent years, the study of lithic raw materials used by prehistoric hunter-gatherers for the
production of stone tools has received much attention. People selected different types of rocks
to influence the technology and type of tools manufactured [1–6]. Such studies can help to
understand the way in which people took advantage of the mechanical attributes of rocks that
affect reduction sequences and edge-wear properties of tools [2, 4, 7]. Sedimentary rocks such
as chert, flint, silicified shale, and other silica rich rocks were commonly used, and this is pre-
sumed to be due to their predictable fracturing properties and good knapping qualities. Raw
material quality is commonly related to the mineralogical structure (e.g., grain size and shape)
and purity of a given material. In most cases, raw materials that have isotropic mechanical
properties are often considered to be of higher quality for tool making [1]. Microstructural
characteristics are also thought to potentially affect the size and/or shape of a final knapped
product [2]. This is supported by replicative experiments conducted by contemporary knap-
pers which have demonstrated that higher quality raw material has a direct influence on the
manufacturing process [8–15].

Some scholars argue that a good raw material that is suitable for knapping should be brittle,
elastic, and isotropic [9, 10]. However, only few semi-quantitative and quantitative studies of
the mechanical properties of lithic materials have been carried out, and these date to the mid-
20th century. Goodman's [16] experimental studies on this subject were among the first to ana-
lyse the hardness, toughness, and density of archaeological stone tools. Every stone tool can be
seen as a unique material that has different raw material structure, morphology, and composi-
tion. The study by Moník and Hadraba [17], attempted to answer the question of whether dif-
ferences in raw material procurement may have been driven by the mechanical properties of
selected materials or not. Studies conducted by Lerner et al. [18] suggest that the physical prop-
erties of lithic raw materials have direct implications for use-wear accrual rates, meaning that
roughness is linked to the rate of wear on stone tools. However, similar works in mechanical
characterization of raw materials from archaeological contexts are still scarce. Some of the
available studies concern mineralogical, chemical and crystallographic transformations [19]. A
few attempted to determine the thermal evolution of fracture mechanics [20–23]. The impor-
tance of the studies of mechanical properties become more pronounced due to the growing
body of evidence that indicates hominins purposefully altered lithic raw materials to increase
the knapping qualities of rocks. This has implications for the evolution of human cognition
[20, 22–26]. For instance, deliberate heat treatment of rocks to alter their knapping quality is
considered a transformative technique [27] and has in the past been used to make inferences
about prehistoric hominin cognition [28].

The current work is based on a systematic investigation of archaeological stone tools and
geological samples of raw materials to determine the mechanical properties from different
Upper Palaeolithic sites of Kazakhstan (Fig 1). Absolute chronology of Upper Palaeolithic sites
is only available for Maibulaq (47–21 ka cal BP) and Ushbulaq (45–39 ka cal BP) [29, 30]. The
chronology of the remaining sites was determined by techno-typological characteristics of the
lithic assemblage. The primary objectives are (1) to test geological and archaeological samples
to determine their mechanical properties, and (2) to preliminarily assess how these properties
affect the knapping technology. Previous studies have been primarily concentrated on silica
materials such as chert and silcrete [18, 20, 23], but other types of sedimentary and volcanic
rocks found in archaeological contexts have received less attention. Here, we attempt to correct
this imbalance by testing both sedimentary and volcanic rocks used by prehistoric people.

In this paper, we analyse chert, shale, and porphyry using indentation testing to investigate
three mechanical properties: fracture resistance, elasticity (also known as Young's modulus),

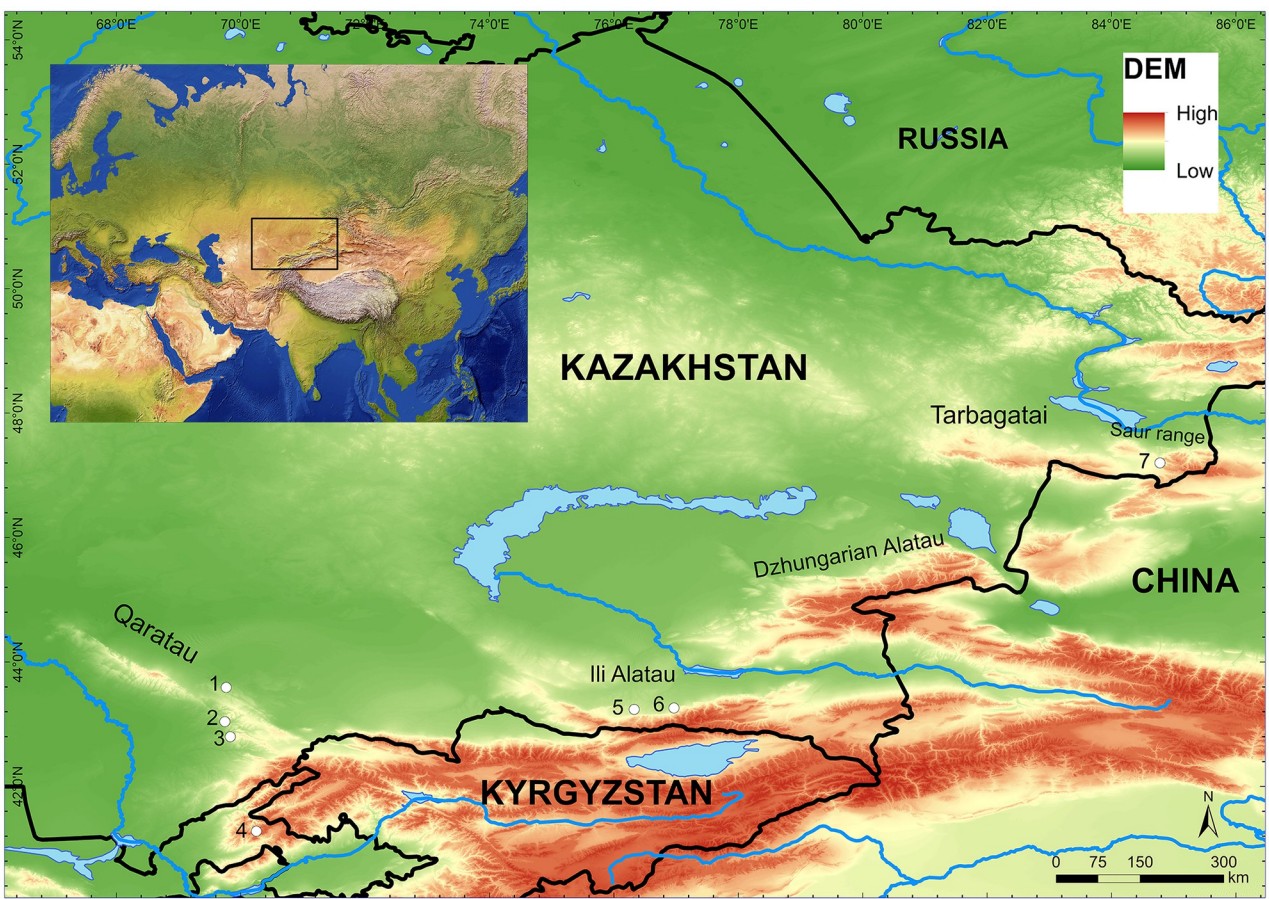

**Fig 1. Palaeolithic sites mentioned in the text and illustrated in relation to the major topography of the Inner Asian Mountain Corridor (Kazakh portion).** 1) Yntaly, 2) Usiktas (surface site), 3) Valikhanova, 4) Kattasai, Uzbekistan, 5) Maibulaq, 6) Rahat, 7) Ushbulaq. Data sources: Global Administrative areas (GADM) [59], vector and raster map data from Natural Earth (www.naturalearthdata.com) and Shuttle Radar Topography Mission (SRTM) Version 4 [60].

and hardness (Table 1) [23, 31]. Additionally, we carry out traditional petrographic analysis to assess the mineralogical composition, impurities within the matrix, and individual grain size of minerals that could potentially influence the fracture behaviour of the studied specimens.

## 2 Regional setting

The materials studied here come from archaeological sites located in the piedmont zones of the Qaratau, Ili Alatau, and the Altai-Tarbagatai mountain ranges located in southern, south-eastern, and eastern Kazakhstan, respectively (Fig 1). It includes much of the Inner Asian Mountain Corridor (henceforth IAMC) [32]. The IAMC is a chain of mountain ranges approximately 2000 km in length in the centre of the Eurasian continent and stretching across most of the Central Asian countries, western China, and Mongolia. Although geologically varied, the study areas share common characteristics of geological formation. Topographic expressions originate from the tectonic activity, erosion, and other depositional processes [33–35]. The mountain ranges under study are all affected by a fault system that separates them into different sub-ranges, i.e., the Greater and Lesser Qaratau [33].

Many Palaeolithic sites are found in the Qaratau range [36]. Its structural and geological settings have been previously characterised by a number of geologists [33–35, 37]. The range

**Table 1. Description of analysed samples.**

| Sample ID | Rock type | Location | Sample type | References |
|---|---|---|---|---|
| UBD-1-20* | Dark shale | Ushbulaq, East Kazakhstan | geological | [58] |
| UBG-1-20* | Green shale | Ushbulaq, East Kazakhstan | geological | [58] |
| YNT-1-20* | Chert | Yntaly, South Kazakhstan | geological | [58] |
| MB-1-20* | Porphyry | Maibulaq, South Kazakhstan | archaeological | [29] |
| UT-22* | Chert | Usiktas, South Kazakhstan | archaeological | [47] |
| UT-144* | Chert | Usiktas, South Kazakhstan | archaeological | [47] |
| UB-526* | Dark shale | Ushbulaq, East Kazakhstan | archaeological | [56] |
| UB-537* | Green shale | Ushbulaq, East Kazakhstan | archaeological | [56] |
| UB-571 | Green shale | Ushbulaq, East Kazakhstan | archaeological | [56] |
| UB-532 | Dark shale | Ushbulaq, East Kazakhstan | archaeological | [56] |
| UB-492 | Green shale | Ushbulaq, East Kazakhstan | archaeological | [56] |
| UB-514 | Dark shale | Ushbulaq, East Kazakhstan | archaeological | [56] |
| UB-616 | Limestone | Ushbulaq, East Kazakhstan | archaeological | [56] |
| UT-48 | Chert | Usiktas, South Kazakhstan | archaeological | [47] |
| UT-10 | Chert | Usiktas, South Kazakhstan | archaeological | [47] |
| UT-181 | Chert | Usiktas, South Kazakhstan | archaeological | [47] |
| UT-217 | Chert | Usiktas, South Kazakhstan | archaeological | [47] |

Localities shown in the table are illustrated on Fig 1. Samples marked with (*) were tested to determine their mechanical properties, while the rest were characterised petrographically.

mainly consists of Neoproterozoic and Palaeozoic bedrocks, and several carbonate seamounts developed due to thermal subsidence of the newly formed crust. The major carbonate platform formed during the Famennian and early Pennsylvanian [33]. The formation of the carbonate platform affected the structure of the Qaratau range, which is mainly composed of limestone, by creating a precondition to form caves, rockshelters, and silica rich rocks within carbonate beds. Such environmental factors played an important role in the human occupation of the region.

Unlike the Qaratau range, the Ili Alatau mountain range (Kazakh portion of the northern Tian Shan) is characterised by steep slopes and the presence of glaciers at higher elevations [34]. The northern foothills enclose the vast depression of the Ili and Dzungarian Alatau to the north-east. The mountain foothills are blanketed of different types of sediments, of which loess covers most of the area. The loess blanket known as the Central Asian piedmont that extends from the Pamir and Alai to the Ili Alatau was extensively studied and defined by Fitzsimmons et al. [29].

These mountain ranges have an arid to semi-arid climate with high variability in temperature variations between different seasons. The position of the mountain groups has played an important role in the early hominin dispersal across Asia [36, 38–40], and possibly provided a refugium during colder episodes of the Pleistocene [41–43]. Archaeological reports indicate extensive human activity, and earlier studies concerning the Palaeolithic of Kazakhstan already revealed that this region has great potential for studying the patterns of human behaviour and migration throughout Central Asia [29, 36, 44–57].

Despite the limited number of studies on raw material, recent work by Namen et al. [58] describes its distribution in the piedmont zones of Kazakhstan. In that paper, we discussed the geographic use patterns of different raw materials using the Centre for Russian and Central EurAsian Mineral Studies (CERCAMS) database. According to macroscopic observations of lithic assemblages, every stratified site has a distinctive type of raw material, probably

outcropping locally in close vicinity to the sites. Due to complex geological formations of the piedmont and foothill zones of the IAMC, this region offers a large amount of knappable raw materials. Therefore, the Palaeolithic sites across Kazakhstan have assemblages knapped on various lithologies. Our work showcases these differences through its comprehensive investigation of different rocks and offers a detailed insight into their mechanical properties. Without understanding the limitations that each raw material imposes on the knapper, our comparison of sites utilising different lithologies might be limited.

## 3 Materials and methods

### 3.1 Samples and sample preparation

For the current study, we selected a total of 17 samples, 14 of which come from archaeological sites, while the remaining three are geological raw materials (Table 1). The geological samples were collected during the PALAEOSILKROAD project's field campaign conducted in 2018 and 2019 [58, 61] under license No. 15008746 (12.05.2015) of the National Museum of the Republic of Kazakhstan based on a collaboration protocol between the Eberhard-Karls University of Tübingen and the National Museum. Primarily, chert and shale outcrops were surveyed and sampled because these materials were commonly used to produce stone tools at different archaeological sites in the piedmont and foothill zones of the IAMC [58]. The archaeological samples come from the stratified sites of Ushbulaq (Eastern Kazakhstan) [30, 56], Maibulaq (Almaty region) [29, 62, 63] and a surface site at Usiktas (South Kazakhstan) [64]. We sampled only lithics from the surface, so as not to disturb the integrity of the assemblages found in stratified contexts. The major sampling criterion was the size of the lithic pieces. Collecting larger samples (approx. 3–5 cm) allowed us to cut the pieces in the middle and then conduct experiments on polished surface produced from the lithics. Chert can be frequently found in stratified and surface sites throughout southern Kazakhstan, and shale is a major raw material of stone tool assemblages from Ushbulaq. Porphyry was commonly utilised in stratified sites at Maibulaq and Rahat [65]. No geological sample of porphyry was included in the current study due to the lack of systematic raw material survey at Maibulaq. We refer to porphyry as a rock of volcanic origin with crystals visible to the naked eye set in a fine grained matrix. The sample descriptions are summarized on Table 1. Location of the sites are shown in Fig 1.

### 3.2 The indentation tests

A total of eight samples (five archaeological and three geological) were tested to determine their mechanical properties. In order to conduct the analyses, plane-parallel plates measuring approximately $40 \times 30 \times 3$ mm (thickness of each sample varies) were cut and diamond polished (see Fig 2) on one side. They were then analysed by the Vickers and Knoop indention [23, 66]. The Vickers and Knoop indentation method is commonly used in material science to measure hardness of materials. The Vickers diamond creates square-shaped indentations, whereas the Knoop diamond produces elongated diamond shaped indents.

Vickers indentations were conducted because it allows to assess data on indentation fracture resistance (according to the method proposed by [31]), Young's modulus (following the protocol proposed by Ben Ghorbal et al. [66]) and hardness of the samples were determined using Knoop indentations. Indentation tests were performed using an Instron 4502 universal testing machine at the laboratory for Applied Mineralogy at Tübingen University's Geosciences department, Germany. Each sample was indented 20 times for each type of indenter to obtain statistically relevant data (see Fig 2). The size of indentations and cracks was determined using images acquired with a HITACHI Tabletop scanning electron microscope (SEM) TM3030. The protocol used was as follows. Load was set to 100 N (with a pre-load of 10 N),

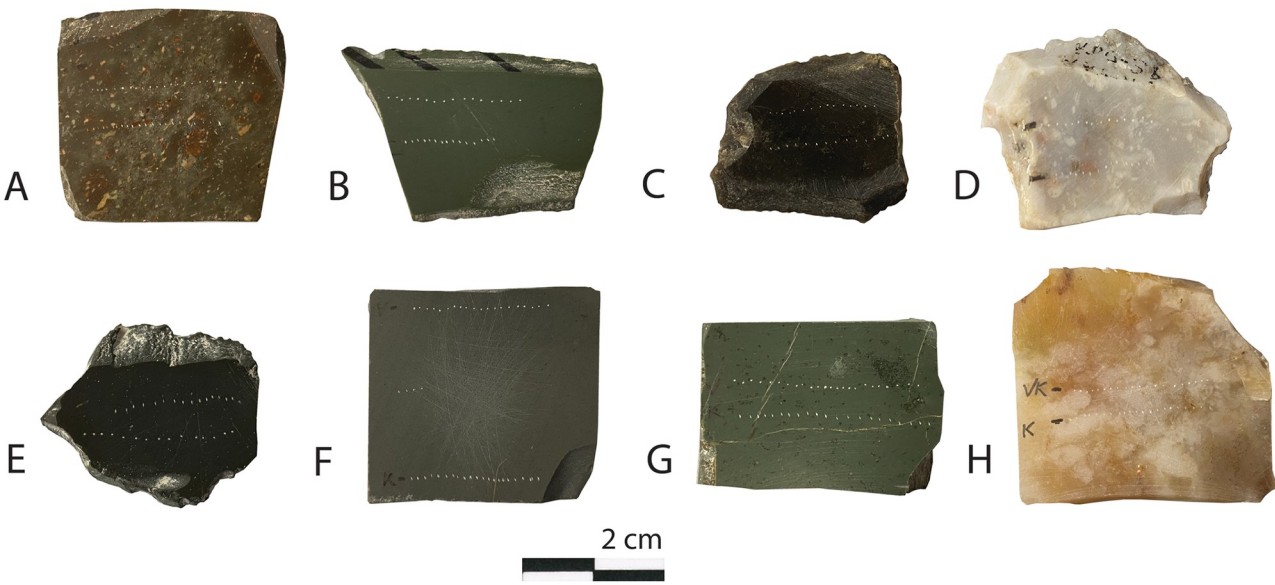

**Fig 2. Illustration of archaeological (A-E) and geological (F-H) samples that undergone mechanical testing by means of indentation tests.** The dotted lines are the impressions of the diamond indents.

speed of indentation 1 mm/s and a hold time of 20 s. The size of each indentation is proportional to material hardness according to Eq (1). Another consequence of the indentation of brittle materials with Vickers and Knoop indenters is the formation of half penny-shaped cracks that can be observed at the polished surface. Cracks apparently depart from all four corners of Vickers indentations and from the two acute angle corners of Knoop indentations. The length of the cracks forming from Vickers indentations are proportional to the value of indentation fracture resistance (a value closely related to fracture toughness (see [67]).

*Knoop hardness* (HK) was calculated by using the formula suggested by Ben Ghorbal et al. [66]. They proposed a way to calculate Knoop hardness that yields results equivalent to the Vickers hardness (equation 12 in Ben Ghorbal et al. [66]):

$$HK = \frac{P}{\frac{b'}{2} \cdot L'}$$
(1)

where,
*P* is the applied load in kilogram-force (KgF);
*b'* is the short diagonal of the Knoop indent in mm;
*L'* is the long diagonal of the Knoop indent in mm.

*Modulus of elasticity* (E) is normally obtained by measuring the potential of a material to deform upon applied stress. However, as bending tests lied outside of the possibilities for this study, we used Ben Ghorbal et al. [66]'s formula that relies on comparing the long and short diagonals of the Knoop indentations. The calculation was initially suggested by Marshall et al. [68], but further refined by Ben Ghorbal et al. [66]:

$$E = 0.417 \times HK \div \left( \frac{1}{7.11} - \frac{b'}{L'} \right)$$
(2)

where,
*HK* is the Knoop hardness in gigapascal (GPa);

*b'* is the short diagonal of the Knoop indent in μm;

*L'* is the long diagonal of the Knoop indent in μm.

*Indentation fracture resistance ($K_{Ic}$)*, as defined by Danzer et al. [67], a value that is closely related to fracture toughness, a measure of a material's resistance to fracture propagation. We used Niihara et al. [31]'s formula to obtain a value for indentation fracture resistance *($K_{Ic}$)*, which uses the length of the surface cracks that develop from the four apical points of the Vickers indentation:

$$K_{Ic} = 0{,}067 H \sqrt{a} \left(\frac{E}{H}\right)^{0,4} \left(\frac{c}{a}\right)^{-1,5} \tag{3}$$

where,

*E* is Young's modulus in megapascal (MPa);

*H* is Vickers hardness in MPa (we used VH here, as calculated from Ben Ghorbal et al. [66], as the differences between HV and HK are negligible when using their calculations);

*c* is the length of the cracks in μm;

*a* is the diagonal of the Vickers indentation in μm.

For this calculation, we admitted that the Knoop hardness is equal to the Vickers hardness and use the value obtained by Eq (1). The size of some of our Vickers indentations could not be accurately determined because their edges flaked off during the experiments. We admitted a standard value of quartz that is equal to 11.65 GPa. Diagonals of those indentations with intact margins were measured.

## 3.3 Petrographic study

A total of 16 thin sections, consisting of 13 archaeological samples from surface collections and 3 geological reference samples were microscopically examined (Table 1). We assessed potential impurities, mineralogical composition, the grain size of quartz grains and relative percentages of clasts and matrix. Traditional petrographic analysis was conducted at the laboratory for Geoarchaeology of the Institute for Archaeological Sciences (INA), University of Tübingen, Germany. The geological and archaeological (surface lithics) samples were prepared following a standard thin section preparation procedure. The thickness of the samples was reduced on a thin section grinder until the thickness of ca. 3 μm reached. Thin sections were analysed using a Zeiss petrographic microscope and photomicrographs were obtained using Axio camera coupled to the microscope. Identification and description of minerals were made under plane polarized light (PPL) and cross-polarized light (XPL) following the terminology outlined by Courty et al. [70] and Stoops [71].

Thin sections of samples tested for mechanical properties were analysed employing the Image J software to calculate the percentages of inclusions and matrix which were estimated using the point counting technique [72]. We aimed to identify these components to evaluate whether there is a relationship with mechanical properties or not.

## 4 Results

Three out of eight samples yielded mechanical values because of the flaking off of all margins which made it impossible to measure indentation diagonals or crack lengths for the other samples. Since our results rely on few successful indentations, the results of the whole scheme must be regarded with some caution. Here, we present the results obtained from chert, shale, and porphyry as well as describe their petrographic features. Values of hardness, elastic modulus, and indentation fracture resistance are summarised below in Tables 2 and 3 and graphically shown in Figs 3–5. Photomicrographs of studied thin sections are shown in Figs 6 and 7.

**Table 2. Hardness (HK), elasticity (E), and fracture resistance (KIc) values as calculated from the indentation tests.**

| Sample ID | Rock type | *HK* | *E* | $K_{Ic}$ |
|---|---|---|---|---|
| UT-22 | Chert | 10.153 | 77 | 2.4 |
| UB-537 | Shale | 4.392 | 153 | 3.01 |
| MB-1-20 | Porphyry | 6.730 | 55 | 1.6 |

Note: the values of *HV* and *HK*, and *E* are given in GPa, and $K_{Ic}$ is given in MPa m$^{1/2}$.

**Table 3. The values of hardness (HK), elastic modulus (E), fracture resistance (KIc), crack lengths of Vickers and Knoops diamonds as calculated from the Vickers and Knoop indentation.**

| Sample name | Crack length of Vickers diamond | | Crack length of Knoop diamond | | HK | E | $K_{Ic}$ |
|---|---|---|---|---|---|---|---|
| | c [μm] | a [μm] | L' [μm] | b' [μm] | [GPa] | [MPa] | [MPa m$^{1/2}$] |
| UT-22 (Chert) | 188 | 125 | 495 | 36 | 11.179 | 68724 | 2.38 |
| UT-22 (Chert) | 188 | 125 | 483 | 44 | 9.283 | 80041 | 2.27 |
| UT-22 (Chert) | 188 | 125 | 487 | 37 | 10.984 | 71581 | 2.40 |
| UT-22 (Chert) | 188 | 125 | 480 | 34 | 12.093 | 73124 | 2.56 |
| UT-22 (Chert) | 188 | 125 | 489 | 41 | 9.786 | 73606 | 2.26 |
| UT-22 (Chert) | 188 | 125 | 470 | 37 | 11.411 | 77487 | 2.53 |
| UT-22 (Chert) | 188 | 125 | 488 | 48 | 8.368 | 86450 | 2.20 |
| UT-22 (Chert) | 188 | 125 | 480 | 44 | 9.418 | 80573 | 2.29 |
| UT-22 (Chert) | | | 476 | 43 | 9.644 | 81872 | |
| UT-22 (Chert) | | | 483 | 44 | 9.363 | 79105 | |
| UB-537 (Shale) | 139 | 125 | 590 | 77 | 4.386 | 188498 | 3.17 |
| UB-537 (Shale) | 132 | 125 | 603 | 75 | 4.397 | 117266 | 2.84 |
| UB-537 (Shale) | 122 | 125 | | | | | |
| UB-537 (Shale) | 136 | 125 | | | | | |
| UB-537 (Shale) | 122 | 125 | | | | | |
| UB-537 (Shale) | 137 | 125 | | | | | |
| UB-537 (Shale) | 137 | 125 | | | | | |
| UB-537 (Shale) | 156 | 125 | | | | | |
| MB-1-20 (Porphyry) | 200 | 125 | 551 | 49 | 7.298 | 60380 | 1.59 |
| MB-1-20 (Porphyry) | 177 | 125 | 601 | 53 | 6.162 | 50410 | 1.61 |
| MB-1-20 (Porphyry) | 204 | 125 | | | | | |
| MB-1-20 (Porphyry) | 182 | 125 | | | | | |
| MB-1-20 (Porphyry) | 199 | 125 | | | | | |
| MB-1-20 (Porphyry) | 195 | 125 | | | | | |
| MB-1-20 (Porphyry) | 102 | 125 | | | | | |
| MB-1-20 (Porphyry) | 182 | 125 | | | | | |
| MB-1-20 (Porphyry) | 155 | 125 | | | | | |
| MB-1-20 (Porphyry) | 143 | 125 | | | | | |
| MB-1-20 (Porphyry) | 147 | 125 | | | | | |
| MB-1-20 (Porphyry) | 176 | 125 | | | | | |

Note that the diagonal value (*a*) was admitted from the standard quartz diagonal due to the flaking off of the edges. It was also admitted as a standard diagonal for the rest of the indented samples.

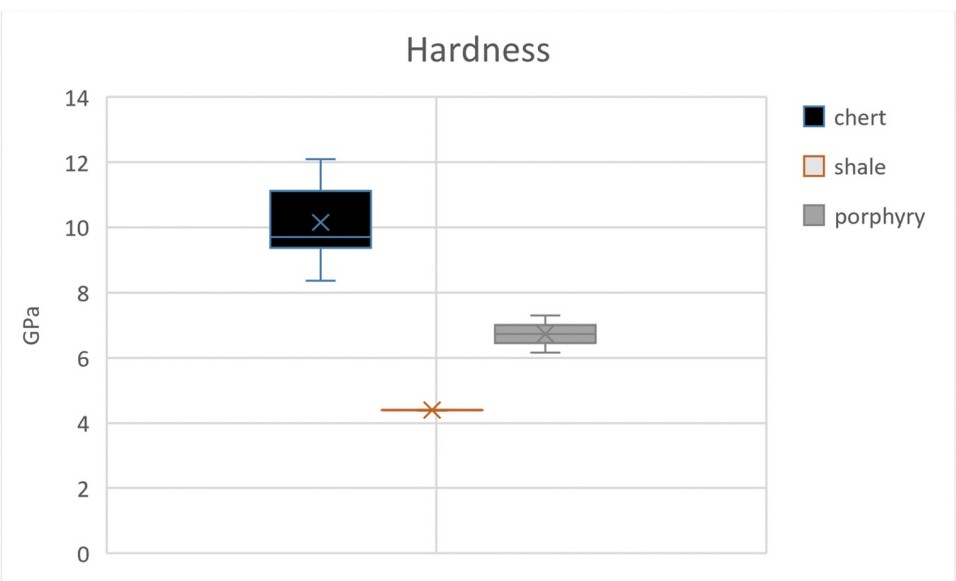

**Fig 3. The hardness values.** The box plot illustrates the highest hardness value for chert and lowest value for shale samples.

## 4.1 Indentation tests

Among the three different types of lithologies studied, chert (UT-22) shows the highest hardness value of over 10 GPa compared to 6.730 GPa for porphyry (MB-1-20) and 4.392 GPa for shale (UB-537). Based on these experiment results, the hardness of shale is the lowest as compared to the other two rocks.

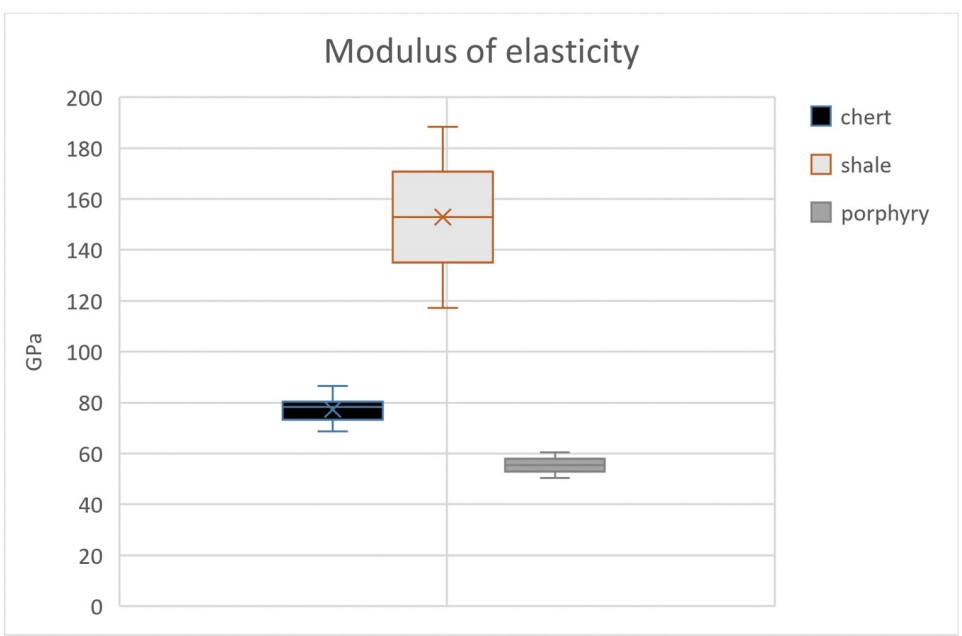

**Fig 4. The boxplot illustrates the modulus of elasticity; values show that shale is less stiff than porphyry and chert.**

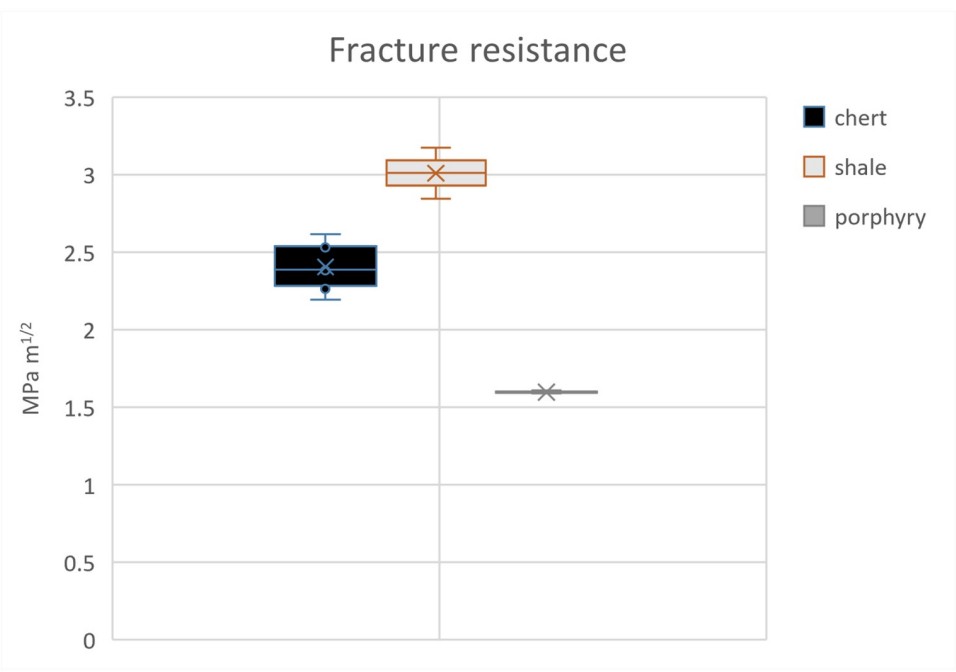

**Fig 5. The fracture resistance of studied samples shown in MPa m$^{1/2}$.** The porphyry sample shows the lowest value (1.5 MPa m$^{1/2}$) as compared to chert samples.

The modulus of elasticity, calculated using Eq (2), suggests that porphyry is the least stiff material with a mean value equal to 55 GPa. While chert yielded a value of 77 GPa, and shale is the stiffest rock among the studied lithologies with an *E* value of 153 GPa. Although porphyry is a volcanic rock that contains relatively large phenocrysts of quartz and feldspar, our results suggest that it is less stiff than the other two analysed samples.

The $K_{Ic}$ values suggest that shale is most resistant to fracture with a mean value of 3.01 MPa m$^{1/2}$. The samples of chert and porphyry have a value equal to 2.4 and 1.6 MPa m$^{1/2}$, respectively. These data may suggest that among three lithologies fracture propagates consuming the least amount of energy in porphyry as compared to shale and chert. However, more detailed

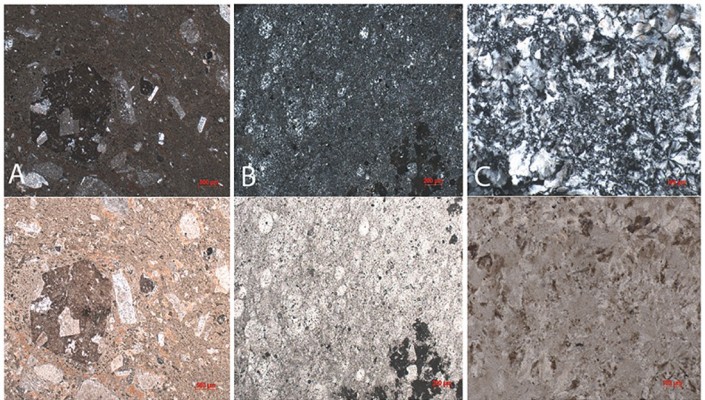

**Fig 6. Photomicrographs of A) porphyry, B) shale, and C) chert under XPL (upper) and PPL (lower) lights that were tested by means of indentation.**

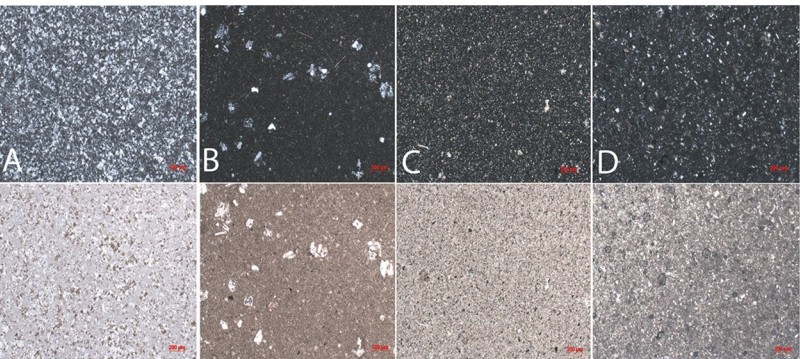

**Fig 7. Photomicrographs of A) chert, B) calcareous shale, C-D) and siliceous shale under XPL (upper) and PPL (lower) lights.**

investigation of rock strength is required to determine whether lower $K_{Ic}$ value equals easier flake detachment.

## 4.2 Petrography

Table 4 summarizes the petrographic characteristics of the studied rock samples. During petrographic analysis, two types of shales were distinguished: calcareous shale (e.g., UBG-1-20) and siliceous shale (e.g., UBD-1-20). The calcareous shale is mainly comprised of well-sorted, subrounded to rounded crystals of calcite and quartz supported by a calcareous matrix. Calcite grains within shale are of secondary origin, which were recrystallised during later rock diagenesis. The siliceous shale consists of subrounded grains of feldspar and quartz. Thin sections

**Table 4. Basic petrographic description of archaeological and geological thin sections.**

| Sample ID | Rock type | Matrix | Description |
|---|---|---|---|
| MB-1-20 | Porphyry (Rhyolite?) | Siliceous | Coarse and angular grains of quartz, feldspar, and sericite, a product of hydrothermal alteration of feldspar, minerals embedded in a fine-grained siliceous matrix. Large xenolith is observed. |
| UBD-1-20 | Dark shale | Calcareous | Fine-grained, rounded and subrounded grains of quartz and calcite are supported by a calcareous matrix. |
| UBG-1-20 | Green shale | Siliceous | Moderately sorted, subrounded grains of quartz and feldspar, and fine-grained calcite minerals are supported by a siliceous matrix. |
| YNT-1-20 | Chert | Siliceous | Entirely composed of length-fast chalcedony |
| UB-537 | Dark shale | Siliceous | Rounded, microcrystalline quartz grains embedded in a siliceous matrix. |
| UB-492 | Green shale | Siliceous | Subrounded grains of quartz and calcite supported by a siliceous matrix. |
| UT-48 | Chert | Siliceous | Entirely composed of length-fast chalcedony. |
| UT-10 | Chert | Siliceous | Entirely composed of length-fast chalcedony. |
| UT-181 | Chert | Siliceous | Entirely composed of length-fast chalcedony. |
| UT-144 | Chert | Siliceous | Entirely composed of length-fast chalcedony. |
| UT-22 | Chert | Siliceous | Entirely composed of length-fast chalcedony. |
| UT-217 | Chert | Siliceous | Entirely composed of length-fast chalcedony. |
| UB-571 | Green shale | Siliceous | Primarily composed of well-rounded, well-sorted quartz grains, few inclusions of angular feldspar can be observed. |
| UB-532 | Dark shale | Siliceous | Composed of primarily well-sorted calcite and few quartz minerals, several internal cracks are filled with calcite. |
| UB-514 | Dark shale | Siliceous | Subrounded grains of clay sized quartz supported on a siliceous matrix. |
| UB-616 | Silicified limestone (?) | Calcareous | Silicified limestone. |

**Table 5. Grain size of quartz minerals and percentage of inclusions and matrix.** CC stands for cryptocrystalline grain size (<1 μm). Note: samples marked with (*) were tested to determine their mechanical properties.

| Sample ID | Rock type | % inclusions | %matrix | Quartz | | |
|---|---|---|---|---|---|---|
| | | | | Min (μm) | Max (μm) | Mean (μm) |
| UBD-1-20 | Dark shale | 34.7 | 55.1 | 146.6 | 566.2 | 299.5 |
| UBG-1-20 | Green shale | 10.2 | 89.8 | 77.2 | 740.8 | 386.9 |
| MB-1-20* | Porphyry | 32.9 | 60.6 | 409.5 | 2241.9 | 1125.4 |
| UB-537* | Shale | 55.1 | 61.3 | 25.1 | 89.3 | 41.9 |
| UT-22* | Chert | CC | CC | CC | CC | CC |

prepared from archaeological lithics from Ushbulaq ($n = 6$) show petrographic characteristics similar to the geological samples with varying degrees of quartz, calcite, and feldspar inclusions with the exception of sample UB-616 (Fig 7).

The thin sections of archaeological lithics from Usiktas ($n = 6$) are composed entirely of length-fast chalcedony, similar to the geological sample YNT-1-20 (Yntaly chert). Microscopically, these samples have similar microstructure. However, we can not safely affirm the transport of these lithic raw materials from Yntaly, we only note their structural similarity. The presence of impurities and other minerals within chert samples is not observed. A thin section of porphyry from Maibulaq (MB-1-20) is comprised of coarse and angular grains of quartz and feldspar supported by a siliceous matrix. An inclusion of xenoliths, possibly broken off from the magma chamber or conduit walls around the time of eruption was observed (Fig 6A).

In addition, individual grain sizes of quartz, percentage of inclusions, and matrix of indented samples (see subchapter 4.1 and Table 1) has been measured and shown in Table 5. This is done to examine whether petrographic characteristics (grain size, number of inclusions, and matrix) affects the fracture resistance of tested rocks. As expected, porphyry contains the largest grain size among other studied samples. These clasts are also visible to the naked eye. The geological samples of shale contain relatively large mineral grains with a mean size equal to 299.5 μm, whereas the grain size of archaeological lithic is much smaller and equals to 41.9 μm. Average grain size of chert could not be measured due to the cryptocrystalline nature of quartz grains in chalcedony.

## 5 Discussion

This is the first experimental work to investigate the mechanical properties of chert, shale, and porphyry from the archaeological context of Kazakhstan. However, our discussion relies on few successful indentations (Table 3) and, therefore, the whole scheme must be regarded with some caution. Despite the small size of samples studied, this work offers insights into some of the mechanical properties of chert, shale, and porphyry which add to the knowledge of knapping technology and the formation of assemblages of stone tools. The results suggest that the raw materials considered to be of lower quality (i.e., porphyry) due to the presence of large phenocrysts have, in fact, mechanical properties (e.g., fracture resistance and modulus of elasticity) that can be compared to materials considered to be of higher quality, i.e., chert and silcrete (studied by Schmidt et al. [23]).

### 5.1 Raw material quality

The studied mechanical properties allow us to have a preliminary discussion on the *meaning of quality* with respect to lithic raw materials. Our results suggest that chert is the hardest

material, harder than porphyry and shale. This is expected since it is entirely composed of length-fast chalcedony, which in turn is quartz (with an average hardness of ∼12 GPa). On the other hand, shale has a lower hardness value despite its siliceous matrix and composition primarily of quartz, as seen under microscope. Although impurities within the shale were not observed during our petrographic study, the lower overall hardness of the rock could be due to the presence of clay impurities within the shale microstructure which were not detectable under 40x magnification [69]. The $E$ values obtained from our experiments likewise show significant differences among chert, shale, and porphyry. Even though chert is composed of chalcedony and shale consists primarily of microcrystalline quartz, the elasticity values of chert and shale differ greatly, too. On the other hand, the $E$ values of porphyry are lower than those of chert and shale. In the analysis of mechanical properties of heat-treated silcrete, Schmidt et al. [23] observed that silcretes with more (abundant) clasts generally had a strong loss of $E$ values upon heat treatment compared to finer silcrete. This means that structural differences developed during the process of heat treatment may affect the elasticity value [23]. This is also supported by our observations on the relationship of grain size and mechanical properties of the rocks. For instance, our investigation shows that samples with larger grain size such as porphyry have a lower modulus of elasticity. Also, based on the measurements using the point counting technique of grain size, inclusions, and matrix, samples with lower percentages of each component demonstrate lower values of indentation fracture resistance (Table 5). An experimental investigation by Huang and Wang [73] on the impact of petrological characteristics, particularly of grain size to fracture toughness, suggests the dependence of these two parameters. However, due to the small size and unheated nature of the studied samples compared to the thermally altered silcretes, our results should be considered with some caution. Further investigations with a larger number of samples are needed to assess whether the large grain size affected the mechanical properties of porphyry or not.

Generally, the modulus of elasticity is proportional to the stiffness of the material. Domanski et al. [20] state that the modulus of elasticity is an important measure of the suitability of materials for blade manufacture because stiffness-controlled fracture propagation is largely responsible for blade detachment [10].

The mean values of $K_{Ic}$ have a linear correlation with the $E$ values, meaning that samples with the lowest $E$ values also present low $K_{Ic}$ values and vice versa (see Figs 3 and 4). The lower values of $K_{Ic}$ imply less resistance against fracture, suggesting that the material is easier to fracture, and perhaps also to knap tools. Thus, the obtained values of $K_{Ic}$ suggests good knapping quality of porphyry, despite the presence of large phenocrysts (see section 5.2 for discussion). Also, the modulus of elasticity and fracture resistance values of porphyry (see Table 2) can be compared to those of chert and silcrete obtained by Schmidt et al. [23]. However, the meaning of $K_{Ic}$ for the actual knapping properties observed during experimental knapping should be investigated in the future.

This is the first time that such experiments were conducted to investigate the mechanical properties of *archaeological* porphyry. Our results offer a preliminary insight into the knapping quality of this rock. Further experimental lithic knapping and research on other mechanical properties (e.g., fracture strength) is necessary to examine the factors that affect the flaking of rock samples. Similar research specifically targeting the evolution of mechanical properties of silica rocks upon heat treatment allow us to compare the $K_{Ic}$ values with previously published data and verify the results. For instance, the fracture resistance values of chert and silcrete from a study by Schmidt et al. [23] vary from 1.3 to 1.85 MPa m$^{1/2}$. The resistance values of our samples vary between 1.6 and 2.4 MPa m$^{1/2}$ and therefore support previous work.

## 5.2 Implications for stone tool knapping

When we compare the results of the experimental works with the stone tool knapping of the Upper Palaeolithic sites where our samples were collected, we see a distinct pattern. Generally, homogeneous, isotropic, fine-grained, and silica rich rocks were normally preferred for the manufacture of stone tools. However, evidence for the utilisation of locally available materials such as porphyry, which is usually considered as "lower" quality, is known from the Central Asian stratified Palaeolithic sites of Rahat and Maibulaq in Kazakhstan [62, 65], as well as Kattasai-1 and Kattasai-2 in Uzbekistan [5, 74]. We concentrate our discussion on the quality of porphyry and the major techno-typological similarities between Palaeolithic industries that utilised this rock as a principal raw material.

Since porphyry was locally available as river pebbles, a large proportion of lithics knapped at the aforementioned sites were made on this material. The use of higher quality exogenous raw materials such as chert is common, but occurs in much smaller quantities. Based on a comprehensive literature review, we observed the presence of Levallois or Levallois-like technologies in all sites that utilise porphyry [5, 62, 65, 74]. Given that Levallois is considered to involve a combination of knapping skill *and* good quality materials, its presence in porphyry-rich assemblages is important to consider. Furthermore, the analysis of the Kattasai-1 assemblage (Uzbekistan) revealed several schemes of reduction, including radial, as well as single and double platform parallel reduction [75]. A similar knapping technology was documented at both Maibulaq and Rahat. The successful production of Levallois blades and radial reduction of cores at these sites clearly demonstrates the link between the quality of raw material and strategies of reduction. We hypothesise that these similarities could be linked to the mechanical properties of porphyry. Additionally, our results suggest that porphyry has at least some properties that can be compared with those of chert. However, compared to chert, porphyry is a coarse grained volcanic rock with a siliceous matrix (see Table 4). In coarser grained rocks the fracture propagates as both transgranular and intergranular cracks, thus consuming less energy to detach a flake of a desired morphology [76]. Moreover, the presence of bladelets knapped on both porphyry and shale from Maibulaq demonstrates its suitability for the production of smaller tools. The fracture resistance, elasticity, and hardness values obtained from our mechanical tests also attest to its suitability for the knapping of blades and bladelets. This data allows us to hypothesise that hominins who once occupied these sites had to adapt the stone knapping technology to the quality and availability of resources. In addition to the grain size of porphyry, we observed that cobbles currently available in the Maibulaq stream bed today tend to have higher incidence of large phenocrysts than those encountered in the archaeological collection. Exploring the pattern of grain size with a sample of both raw material and archaeological stone tools could potentially indicate that prehistoric communities at Maibulaq deliberately selected rocks with smaller phenocrysts for knapping. However, this must be tested with further analyses of raw materials and archaeological stone tools.

## 6 Conclusions

This is the first time that mechanical tests including fracture resistance, hardness, and the modulus of elasticity have been conducted on unheated samples of archaeological lithics and unmodified rocks at several Upper Palaeolithic sites in Kazakhstan. Our study offers a first insight into the quality of archaeological porphyry, which was used as a principal raw material at Upper Palaeolithic sites in the piedmont zones of the IAMC. We conclude that raw materials that were previously thought of as lower quality (e.g., porphyry) have some mechanical properties similar to those of chert. The prehistoric knappers that inhabited the northern foothills of the IAMC adapted their lithic reduction schemes to the quality of available raw material such

as porphyry. The presence of Levallois or Levallois-like technology that can be observed from several Upper Palaeolithic sites utilising porphyry as a main raw material further support this hypothesis.

This work highlights the importance of the mechanical properties of chert, shale, and porphyry and their effect on the production of stone tools. Our findings show that materials which we expect to be difficult to knap based on visual inspection may have good fracture properties when evaluated objectively. Porphyry comes as a surprise because it is rarely used for knapping, and mainly unfamiliar to archaeologists [5, 65]. However, these kinds of counterintuitive results are also likely for some materials that *are* familiar to archaeologists, such as different types of cherts, basalts, and other materials.

Further investigations involving larger numbers and various types of raw materials and archaeological lithics are necessary to study and eventually build a library including the mechanical properties of the whole corpus of raw materials available in Kazakhstan. Our preliminary data can serve as a baseline for future quantitative and experimental investigations.

## Acknowledgments

We would like to thank Dean Mendigul Nogaibaeva, Prof. Gani Omarov and Dr. Rinat Zhumatayev (Department of Archaeology, Ethnology and Museology, Faculty of History, Ethnology and Archaeology, Al Farabi Kazakhstan National University) for their support of this work. A. Namen would like to thank Prof. Dr. Christopher Miller for his support of this research, and access to the equipment of the Geoarchaeology laboratory at the Institute for Archaeological Sciences, in the University of Tübingen. We would like to thank the two anonymous reviewers for their constructive and helpful comments that have significantly improved the manuscript.

## Author Contributions

**Conceptualization:** Abay Namen, Radu Iovita, Patrick Schmidt.

**Data curation:** Abay Namen, Patrick Schmidt.

**Formal analysis:** Abay Namen, Patrick Schmidt.

**Funding acquisition:** Radu Iovita.

**Investigation:** Abay Namen, Radu Iovita, Klaus G. Nickel, Aristeidis Varis, Zhaken Taimagambetov, Patrick Schmidt.

**Methodology:** Abay Namen, Klaus G. Nickel, Patrick Schmidt.

**Project administration:** Radu Iovita, Zhaken Taimagambetov.

**Resources:** Radu Iovita, Klaus G. Nickel, Patrick Schmidt.

**Supervision:** Radu Iovita, Klaus G. Nickel, Patrick Schmidt.

**Validation:** Radu Iovita, Klaus G. Nickel, Patrick Schmidt.

**Visualization:** Abay Namen, Patrick Schmidt.

**Writing – original draft:** Abay Namen, Patrick Schmidt.

**Writing – review & editing:** Abay Namen, Radu Iovita, Klaus G. Nickel, Aristeidis Varis, Patrick Schmidt.

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
