## [Decision Letter · Decision Letter 0]

26 Jan 2022

PONE-D-21-32866Mechanical properties of lithic raw materials from Kazakhstan: comparing chert, shale, and porphyryPLOS ONE

Dear Dr. Abay Namen,

Thank you for submitting your manuscript to PLOS ONE. After careful consideration, we feel that it has merit but does not fully meet PLOS ONE’s publication criteria as it currently stands. Therefore, we invite you to submit a revised version of the manuscript that addresses the points raised during the review process.

We look forward to receiving your revised manuscript.

Kind regards,

Enza Elena Spinapolice, Ph.D

Academic Editor

PLOS ONE

Journal Requirements:

2. In your manuscript, please provide additional information regarding the specimens used in your study. Ensure that you have reported specimen numbers and complete repository information, including museum name and geographic location.

For more information on PLOS ONE's requirements for paleontology and archaeology research, see https://journals.plos.org/plosone/s/submission-guidelines#loc-paleontology-and-archaeology-research.

This project has received funding from the European Research Council (ERC) under the European Union’s Horizon 2020 research and innovation programme (grant agreement n˚714842; PALAEOSILKROAD project).

This project has received funding from the European Research Council (ERC) under the European Union's Horizon 2020 research and innovation programme (grant agreement n° 714842; PALAEOSILKROAD project). 

This project has received funding from the European Research Council (ERC) under the European Union’s Horizon 2020 research and innovation programme (grant agreement n˚714842; PALAEOSILKROAD project).

6. We note that Figure 1 in your submission contain [map/satellite] images which may be copyrighted. All PLOS content is published under the Creative Commons Attribution License (CC BY 4.0), which means that the manuscript, images, and Supporting Information files will be freely available online, and any third party is permitted to access, download, copy, distribute, and use these materials in any way, even commercially, with proper attribution. For these reasons, we cannot publish previously copyrighted maps or satellite images created using proprietary data, such as Google software (Google Maps, Street View, and Earth). For more information, see our copyright guidelines: http://journals.plos.org/plosone/s/licenses-and-copyright.

Reviewers' comments:

Reviewer's Responses to Questions

**Comments to the Author**

1. Is the manuscript technically sound, and do the data support the conclusions?

Reviewer #1: Yes

Reviewer #2: Yes

2. Has the statistical analysis been performed appropriately and rigorously? 

Reviewer #1: Yes

Reviewer #2: Yes

3. Have the authors made all data underlying the findings in their manuscript fully available?

Reviewer #1: Yes

Reviewer #2: Yes

4. Is the manuscript presented in an intelligible fashion and written in standard English?

Reviewer #1: No

Reviewer #2: Yes

5. Review Comments to the Author

Reviewer #1: The paper entitled “Mechanical properties of lithic raw materials from Kazakhstan: comparing chert, shale, and porphyry” by Abay Namen and colleagues aim to test some mechanical properties of raw materials from Kazakhstan in a comparative perspective, including petrographic studies as well. The main objectives of the study are:

1- to test geological and archaeological samples to determine their mechanical properties, and 2- to preliminarily assess how these properties affect the knapping technology.

The paper focuses on an interesting topic: the importance of mechanical properties of raw materials and its implication for the comprehension of hominin behavior and lithic technology. Especially in the area of Kazakhstan, where the study has been settled, and in general for Central Asia. Thus, this is a great contribution to the knowledge of the aforementioned area.

The manuscript is well written and structured in clear sections, with many tables, which are very welcomed to understand the methodology of analysis. The data are available, and most of my comments are detailed in the attached pdf. The research background is well illustrated, as well as figures.

I think the experiments and test the authors used are appropriate to the research questions and objectives, and there are enough data to make the study replicable. The paper offers a window to a wider potential research topic that needs to be improved. The results shown in the manuscript highlight the importance of these types of objective tests, which represent an excellent tool to understand and interpret the archaeological contexts and the behavioral aspects related to the groups that occupied them.

For this manuscript, I would recommend minor revisions before publication, based on the following points that require clarification:

1. More details should be given on the collection of samples, the criteria applied to the selection, and the archaeological contexts where they came from (see attached pdf. for comments).

2. The tables presented in the text are cited in a confused order (e.g. Tab 4 is mentioned before tabs 2 and 3).

3. Captions to Figures 3, 4, and 5 can be improved. It would be helpful to name the graph the authors used to illustrate the results of the tests.

Finally, I would recommend the revision of the manuscript by an English native speaker.

Reviewer #2: Comments to Authors:

This paper presents engaging data on raw material from Kazakhstan Upper Paleolithic sites and proposes an interesting methodology to study their mechanical properties. The indentation tests were complemented by traditional petrographic studies, which offer a supportive tool to support their hypothesis, re-evaluating the importance of porphyry as the primary source of artefacts manufacture. The manuscript is interesting, as it adds some details to an appealing – yet little known – topic, namely the exploited raw materials and their properties in central Asia during Upper Palaeolithic. Although the complete set of geological samples is wished to be enlarged in future studies, the described data represent a good overview, despite the absence of a porphyry geological sample that could have perfectly completed the framework.

However, several parts in the paper should be updated and rephrased to make them meet the specific targets of the publication.

The author's hypotheses are quite convincing, but probably more analyses should be necessary in order to adequately and undoubtedly clarify the unanswered question and extend the lithoteque, and thus raw material circulation routes.

At various places, data are presented in a much too concise way (as for the test and archaeological sample description), while the procedural part is sometimes too detailed. I am attaching the manuscript with in-text comments and questions, and you will find below a list of minor comments.

My general and minor comments are discussed hereafter. See the pdf for detailed comments. All the words or periods reported between "" indicate the proposed alternatives to the original text.

All comments and deletions are followed by side-comment. If you are not able to see them, let the editor know it and I'll provide another file.

To conclude, this is a significant and valuable contribution to the archaeology of the region and for the lithic studies, and I recommend its publication with minor-to-moderate revisions.

General points:

1. The 'archaeological' part of the manuscript is absent. More information related to the contexts, chronological framework, and reference sites from where the absolute dates come from is more than welcomed. I do not think this might be considered a big issue – but on the other hand, it is true that the 'archaeological' descriptions of the investigated specimens are way too short.

2. A minor problem is represented by the too "simplistic" lexicon that makes it look like an essay. I suggest a general revision from a native English speaker or a language polishing with special attention for syntax and words repetition. Also, the verbs concordance needs some attention.

3. Lastly, there are some concepts repeated throughout the text. Try to delate these parts and substitute them with a brief explanation of why you chose specific approaches and what they consisted of.

Minor comments:

· Citation of some page numbers in the Bibliography must be standardized.

· The illustration section should be slightly modified, adding details to the pictures of the sampled areas. You will find detailed requests in the pdf.

6. PLOS authors have the option to publish the peer review history of their article (what does this mean?). If published, this will include your full peer review and any attached files.

Reviewer #1: No

Reviewer #2: No

---

## [Author Response · Author response to Decision Letter 0]

1 Feb 2022

Dear Dr. Spinapolice,

Many thanks for your work in managing this process, and for considering our manuscript for publication. In the revised cover letter, we address your questions regarding the possibility of copyrighted satellite imagery in our figures (our figures do not contain satellite imagery, so there is no copyright issue in this regard).

We greatly appreciate of the insightful comments on our manuscript from two anonymous reviewers. We respond to their comments in detail below.

Many thanks.

Dear Reviewer 1,

Thank you for your valuable work, the time you have spent on doing the review, and the comments you provided on our manuscript, especially your recommendations for clarity in phrasing and additional archaeological information. We appreciate your expertise and your constructive suggestions, which have helped us improve the quality of the paper.

We have revised the manuscript and addressed the issues indicated in your report. Changes are tracked in the revised version of the manuscript. Additionally, please find a point-to-point response to your questions/comments below.

Thanks again and kind regards.

# Reviewer 1

The paper entitled “Mechanical properties of lithic raw materials from Kazakhstan: comparing chert, shale, and porphyry” by Abay Namen and colleagues aim to test some mechanical properties of raw materials from Kazakhstan in a comparative perspective, including petrographic studies as well. The main objectives of the study are: 1- to test geological and archaeological samples to determine their mechanical properties, and 2- to preliminarily assess how these properties affect the knapping technology. The paper focuses on an interesting topic: the importance of mechanical properties of raw materials and its implication for the comprehension of hominin behavior and lithic technology. Especially in the area of Kazakhstan, where the study has been settled, and in general for Central Asia. Thus, this is a great contribution to the knowledge of the aforementioned area.

The manuscript is well written and structured in clear sections, with many tables, which are very welcomed to understand the methodology of analysis. The data are available, and most of my comments are detailed in the attached pdf. The research background is well illustrated, as well as figures. 

I think the experiments and test the authors used are appropriate to the research questions and objectives, and there are enough data to make the study replicable. The paper offers a window to a wider potential research topic that needs to be improved. The results shown in the manuscript highlight the importance of these types of objective tests, which represent an excellent tool to understand and interpret the archaeological contexts and the behavioral aspects related to the groups that occupied them. 

For this manuscript, I would recommend minor revisions before publication, based on the following points that require clarification:

1. More details should be given on the collection of samples, the criteria applied to the selection, and the archaeological contexts where they came from (see attached pdf. for comments).

We have added these details in the revised version of the manuscript. 

2. The tables presented in the text are cited in a confused order (e.g. Tab 4 is mentioned before tabs 2 and 3).

We have fixed this issue in the manuscript. Please, see the revised version.

3. Captions to Figures 3, 4, and 5 can be improved. It would be helpful to name the graph the authors used to illustrate the results of the tests.

Thank you for this remark. We have added additional information in the captions of Fig 2-5.

Finally, I would recommend the revision of the manuscript by an English native speaker.

The revised version of the manuscript has been proofread by a native English speaker. 

Dear reviewer 2,

Thank you for your valuable work, the time you have spent on doing the review, and the comments you provided on our manuscript. We appreciate your expertise and your constructive suggestions, which have helped us to improve the quality of the paper.

We have revised the manuscript and addressed the issues indicated in your report. Changes are tracked in the revised version of the manuscript. As well, please find a point-to-point response to your questions/comments below.

Thanks again and kind regards.

# Reviewer 2: 

This paper presents engaging data on raw material from Kazakhstan Upper Paleolithic sites and proposes an interesting methodology to study their mechanical properties. The indentation tests were complemented by traditional petrographic studies, which offer a supportive tool to support their hypothesis, re-evaluating the importance of porphyry as the primary source of artefacts manufacture. The manuscript is interesting, as it adds some details to an appealing – yet little known – topic, namely the exploited raw materials and their properties in central Asia during Upper Palaeolithic. Although the complete set of geological samples is wished to be enlarged in future studies, the described data represent a good overview, despite the absence of a porphyry geological sample that could have perfectly completed the framework. 

However, several parts in the paper should be updated and rephrased to make them meet the specific targets of the publication.

The author's hypotheses are quite convincing, but probably more analyses should be necessary in order to adequately and undoubtedly clarify the unanswered question and extend the lithoteque, and thus raw material circulation routes. 

At various places, data are presented in a much too concise way (as for the test and archaeological sample description), while the procedural part is sometimes too detailed. I am attaching the manuscript with in-text comments and questions, and you will find below a list of minor comments.

My general and minor comments are discussed hereafter. See the pdf for detailed comments. 

All the words or periods reported between "" indicate the proposed alternatives to the original text.

All comments and deletions are followed by side-comment. If you are not able to see them, let the editor know it and I'll provide another file.

To conclude, this is a significant and valuable contribution to the archaeology of the region and for the lithic studies, and I recommend its publication with minor-to-moderate revisions.

General points:

1. The 'archaeological' part of the manuscript is absent. More information related to the contexts, chronological framework, and reference sites from where the absolute dates come from is more than welcomed. I do not think this might be considered a big issue – but on the other hand, it is true that the 'archaeological' descriptions of the investigated specimens are way too short.

This is true. We have improved missing information concerning the chronology and context of the archaeological sites. They are also cited appropriately for further details. 

2. A minor problem is represented by the too "simplistic" lexicon that makes it look like an essay. I suggest a general revision from a native English speaker or a language polishing with special attention for syntax and words repetition. Also, the verbs concordance needs some attention.

Thank you for your remark. The revised version of the manuscript has been proofread by a native English speaker. 

3. Lastly, there are some concepts repeated throughout the text. Try to delate these parts and substitute them with a brief explanation of why you chose specific approaches and what they consisted of.

We have fixed these issues throughout the manuscript according to your side-comments in the PDF file. 

Minor comments:

· Citation of some page numbers in the Bibliography must be standardized.

This issue was solved. 

· The illustration section should be slightly modified, adding details to the pictures of the sampled areas. You will find detailed requests in the pdf.

Thank you for the remarks. We have incorporated these changes in the revised version of the manuscript.

---

## [Decision Letter · Decision Letter 1]

7 Mar 2022

Mechanical properties of lithic raw materials from Kazakhstan: comparing chert, shale, and porphyry

PONE-D-21-32866R1

Dear Dr. Namen

We’re pleased to inform you that your manuscript has been judged scientifically suitable for publication and will be formally accepted for publication once it meets all outstanding technical requirements.

Kind regards,

Enza Elena Spinapolice, Ph.D

Academic Editor

PLOS ONE

Additional Editor Comments (optional):

Reviewers' comments:

Reviewer's Responses to Questions

**Comments to the Author**

1. If the authors have adequately addressed your comments raised in a previous round of review and you feel that this manuscript is now acceptable for publication, you may indicate that here to bypass the “Comments to the Author” section, enter your conflict of interest statement in the “Confidential to Editor” section, and submit your "Accept" recommendation.

Reviewer #1: All comments have been addressed

Reviewer #2: All comments have been addressed

2. Is the manuscript technically sound, and do the data support the conclusions?

Reviewer #1: Yes

Reviewer #2: Yes

3. Has the statistical analysis been performed appropriately and rigorously? 

Reviewer #1: Yes

Reviewer #2: Yes

4. Have the authors made all data underlying the findings in their manuscript fully available?

Reviewer #1: Yes

Reviewer #2: Yes

5. Is the manuscript presented in an intelligible fashion and written in standard English?

Reviewer #1: Yes

Reviewer #2: Yes

6. Review Comments to the Author

Reviewer #1: The paper by Abay Namen and colleagues has significantly improved. I thank the authors to had taken into consideration my comments and suggestions. I still believe that this paper focuses on a good and interesting topic. I think it is a great contribution to the knowledge of a not well-known area.

The methodology of sample selection is now clear and well explained. As well as some other points I raised on results, discussion, and conclusion sections.

I recommend the paper for publication. I believe the work is well done, nicely described, with all the data and analysis available and understandable.

Reviewer #2: The study presents the results of original research which presents for the first time data coming from an understudied area. Experiments, statistics, and petrographic analyses are described in sufficient detail and the processes are well explained. The references are punctual and were well enriched after the revision.

The only unaccomplished part is related to figure 1, where I warmly recommend enlarging the dots and the numbers identifying the sites (as there is enough space).

In paragraph 5.2, lines 413 414, the author says "...results suggest that porphyry has at least some properties that can be compared with those of chert." Please, specify which properties. It would be nice to have it well stated before the conclusions.

Another minor thing: in Table 1, the first [58] has two right brackets, please delete the typo.

To conclude, the research meets all applicable standards for the ethics of experimentation and research integrity and offers access to the data, as they are reported in proper tables. The conclusions presented are supported by the data, conscious that this is a seminal study that should be continued in the future – and I hope that it will be continued!

This is a significant and valuable contribution to the archaeology of Kazakhstan and raw material studies, and I recommend its publication.

7. PLOS authors have the option to publish the peer review history of their article (what does this mean?). If published, this will include your full peer review and any attached files.

Reviewer #1: No

Reviewer #2: No

---

## [Editor Report · Acceptance letter]

10 Mar 2022

PONE-D-21-32866R1 

Mechanical properties of lithic raw materials from Kazakhstan: comparing chert, shale, and porphyry 

Dear Dr. Namen:

I'm pleased to inform you that your manuscript has been deemed suitable for publication in PLOS ONE. Congratulations! Your manuscript is now with our production department. 

Kind regards, 

on behalf of

Dr. Enza Elena Spinapolice 

Academic Editor

PLOS ONE